# Navigating Neighbourhood Opposition and Climate Change: Feasibility and Acceptability of a Play Street Pilot in Sydney, Australia

**DOI:** 10.3390/ijerph20032476

**Published:** 2023-01-30

**Authors:** Josephine Y. Chau, Putu Novi Arfirsta Dharmayani, Helen Little

**Affiliations:** 1Department of Health Sciences, Faculty of Medicine, Health and Human Sciences, Macquarie University, Macquarie Park, NSW 2109, Australia; 2School of Education, Macquarie University, Macquarie Park, NSW 2109, Australia

**Keywords:** play, physical activity, community, implementation science, evaluation, health promotion

## Abstract

Background: Play Streets are community-led initiatives that provide opportunities for outdoor play and recreation when parks or other facilities may not be easily accessible. This pragmatic evaluation aims to determine the feasibility and acceptability of a pilot Play Street in Inner West Sydney. Methods: We used a post-only mixed methods design. Brief intercept surveys with pilot Play Street visitors assessed their reasons for attending the event and perceptions thereof. Semi-structured interviews explored stakeholders’ experiences related to planning and implementing Play Streets. Results: Approximately 60 adults and children attended the pilot Play Street. The majority of survey respondents (n = 32) were female, aged 35–54, lived in the Play Street’s postcode, and visited in groups consisting of adults and children. Overall respondents rated the pilot positively in enjoyment (100%), safety (97%), and organisation (81%), although there were significant differences between certain demographic subgroups in the perception of organisation and the children’s enjoyment of the pilot Play Street. Stakeholder interviews (n = 2) highlighted the importance of community consultation and reaching compromises, noting concerns about safety and insurance costs, and emphasised the role of Council as a facilitator to help residents take ownership of Play Streets. Delays due to community concerns, poor air quality arising from bushfires, heavy rain on the event day, and COVID-19 lockdowns hindered pilot Play Street implementation and evaluation. Conclusion: This pilot demonstrated that Play Streets are a feasible and acceptable way to use streets as outdoor recreation spaces in Sydney’s Inner West. The evaluation highlights two elements for future sustainability: managing neighbourhood opposition and adapting to climate change.

## 1. Introduction

The important role of play as a key part of every child’s life is acknowledged in the United Nations Convention on the Rights of the Child (UNCRC) [1]. Article 31 supports a child’s right to rest and leisure, and to participate in play and recreational activities appropriate to the age of the child [1]. This acknowledgement in the UNCRC recognises the significant role of play for all aspects of children’s development, health, and wellbeing.

Play, in all its forms, offers a range of benefits for children’s perceptual–motor, cognitive, social—emotional, and language development as well as physical and mental health [2]. Outdoor play, particularly in natural environments, is associated with a range of health benefits including increased physical activity and emotional wellbeing [3]. A systematic review by Brussoni and colleagues [4] found that greater independent mobility where children can play away from direct adult supervision was associated with higher levels of physical activity and sociability.

Despite these benefits, there has been a growing concern that children growing up today have reduced opportunities for play, particularly outdoors [5]. Outdoor environments provide a natural context that satisfy children’s innate curiosity and motivation to explore the world around them. However, a growing body of research conducted over the past two decades highlights that children’s opportunities for outdoor play and independent mobility have decreased both in terms of age at which children are permitted to explore their neighbourhood without adult supervision and the distance from home they are permitted to range [6,7,8,9,10,11]. These studies have identified a number of factors that have impacted children’s freedom to independently explore and be visible within their neighbourhoods including urban density, increased traffic, stranger danger, and less tolerance of adults towards children playing in areas not designated for play (i.e., playgrounds). Consequently, children’s outdoor play is restricted to their home or playgrounds under adult supervision.

Children’s play not only supports their development and wellbeing, opportunities for play also positively impacts their families and others in the community. It provides the context for children and adults to form attachments to others and develop a sense of belonging in their community. When both the environmental (e.g., availability and accessibility of places to play) and social conditions (attitudes towards children playing in public spaces) allow, children will play outdoors. When these conditions exist, play can provide a context for community building, through shared use of space, trust, freedom, and intergenerational interactions [12]. While features of the built environment encourage more physical activity and outdoor play in children [13,14,15], it is often not feasible nor quick to modify the built environment. Play Streets constitute a low-cost reclamation of existing residential spaces for communal use and children’s play [16].

A ‘Play Street’ is an outdoor play space created by temporarily closing off the street to through traffic so that children can engage in active play. Play Streets are community-led initiatives that aim to create a welcoming, inclusive, and safe environment for both parents and children [17]. Previous evaluation studies of Play Streets have demonstrated positive impacts on children’s physical activity and community connections [18,19,20,21]. The evidence shows that Play Streets provide opportunities for safe outdoor play for children with benefits for health and the community [22]. However, there is a relative paucity of evidence about the planning, implementation, and evaluation of Play Streets to understand the factors that would facilitate replication and sustainability thereof. [23,24] Furthermore, Play Streets have synergies with the United Nation’s Sustainable Development Goals, including contributing to good health and wellbeing (SDG 3), sustainable cities and communities (SDG 11), and climate action (SDG 13) [25].

The Inner West Council of Sydney, Australia initiated a Play Streets pilot program as part of its Recreation Strategy, which was designed against a backdrop of increasing population density and limited scope for creating new recreation spaces in the locality [26]. This initiative was part of a suite of projects designed to improve access to outdoor recreation space for Inner West community members after an extensive community needs assessment identified the need for safe and accessible places to play for all ages and abilities as a key priority. Play Streets were considered a way to leverage off existing outdoor space in higher density areas of the council area to provide additional outdoor play opportunities for children and to promote better health and wellbeing across the community. This IWC initiative provided the chance to understand and generate evidence about the implementation of Play Streets in an Australian context. The findings of this study would be significant to practitioners and stakeholders at the intersection of local government, recreation, public health, community, and built environment.

This paper describes the pragmatic evaluation of a pilot Play Street in Inner West Sydney to determine its feasibility and acceptability. The aims of this study were to:(a)Assess community attendance of the pilot Play Street: Who visited the event and what were their characteristics? How long did visitors spend at the event and in what kinds of activities did they engage there?(b)Examine community members’ experiences and perceptions about the event: What were their views and perceptions of the Play Street pilot?(c)Understand stakeholder perspectives about the pilot Play Street: How did community stakeholders feel about holding Play Streets in the Inner West Council area?(d)Identify enablers and barriers to implementation of Play Streets: What factors contributed to the success of the pilot Play Street? What challenges did organisers face in planning and implementing the pilot Play Street?

## 2. Materials and Methods

### 2.1. Study Setting

This study was conducted in the Inner West Local Government Area (LGA) of Sydney, Australia. The process of seeking community interest in Play Streets began in early 2019 with one pilot Play Street held in March 2020.

### 2.2. Study Design

This was a mixed methods evaluation consisting of a cross-sectional survey during the pilot Play Street and stakeholder interviews after the Play Street event was held. The evaluation was designed in consultation with the recreation services team of the Inner West Council.

### 2.3. Participants

Play Street visitors were invited to provide their feedback about the event via brief intercept surveys. Participation was voluntary and anonymous; visitors provided verbal consent to the research team and their responses were recorded digitally using Qualtrics (Provo, UT, USA) on an iPad. Eligible participants were pilot Play Street visitors who were aged at least 18 years old, spoke English well enough to respond to survey questions, and attended the pilot Play Street during the designated event time.

We also conducted semi-structured interviews with stakeholders. Eligible participants were community members and LGA personnel and partners with relevant experience and knowledge of Play Streets (e.g., as part of work role, prior or current experience as participant or organiser, involved in planning Play Streets), aged at least 18 years old and with sufficient English language proficiency.

### 2.4. Play Street Intervention

The pilot Play Street was selected through a nomination process. Residents self-nominated their street to be closed to traffic for children’s play in response to a call from the local council. From 65 nominations, four sites were shortlisted based on five selection criteria: priority was given to streets in areas with lower access to open spaces; the street could be closed with minimal traffic impacts for residents of the nominated street and surrounding areas; the street surface and surroundings were suitable and safe; equitable geographic distribution across the local government area; and having one or more local volunteers willing to organise the Play Street and coordinate with council [27]. A community consultation phase with residents of the shortlisted streets led to three sites withdrawing from the initiative and the identification of one site in the suburb of Leichhardt which met the majority of selection criteria and was deemed suitable for the Play Street trial.

The pilot Play Street was originally set to take place in late January 2020. This date was set following a second extensive consultation process with residents during which the pilot Play Street’s local organising team and champions negotiated event date(s), event duration, and street closure boundaries to accommodate the varying supportive and opposing sentiments from residents about holding a Play Street. The January 2020 Play Street was postponed to March due to poor air quality arising from the Black Summer bushfire season of 2019–2020. The pilot Play Street was finally held on the 15 March 2020 from 3 pm to 5 pm, one week prior to the commencement of the first Sydney COVID-19 lockdown. The weather on the day of the pilot Play Street was poor with heavy rain falling during the morning and early afternoon. The rain cleared after 3 pm and the organisers decided to proceed with the Play Street during the break in the rain.

### 2.5. Measurement

#### 2.5.1. Intercept Survey

The intercept survey asked pilot Play Street visitors about their reasons for visiting the pilot Play Street: how they heard about the event; how they travelled to the event; how long they stayed at the Play Street; whether they have been to a Play Street before; how they rate the pilot Play Street in terms of safety, organisation, and enjoyment; whether other Play Streets should be held in the future; and their children’s experience (if relevant). We also collected broad demographic information: age group, gender, and postcode of residence. The intercept survey was designed based on those used previously in the Sydney Parks Study [28] and Narrabeen Loop Evaluation [29].

#### 2.5.2. Semi-Structured Stakeholder Interviews

We interviewed stakeholders about their perceptions of organising and implementing Play Streets in the Inner West Council area and in other local government areas. The topics covered included: factors that contribute to successful Play Streets; challenges faced or anticipated when organising Play Streets; experience of Play Streets as a visitor and/or organiser; and whether Play Streets would contribute to the local community.

### 2.6. Procedures

The evaluation team carried out the brief intercept surveys with adult visitors during the pilot Play Street. Each survey took about 5–10 min to complete. After the event, an invitation to participate in interviews was sent by the Inner West Council Recreation Officer to their internal stakeholders. The research team also sent an interview invitation to recreation officers and individuals in relevant roles in other Greater Sydney LGAs. Interested individuals contacted the research team for further information. They joined the study by providing written informed consent to the research team. The interviews were conducted via video conferencing (MS Teams) and recorded for transcription purposes.

### 2.7. Data Analysis

Survey data were analysed to provide descriptive statistics about the Play Street visitors, including their demographic characteristics, utilisation, and perceptions of the event. The demographic variables were presented as frequencies (n) and percentages (%). The Independent samples *t*-test was performed to examine the differences in perceptions of Play Street between demographic sub-groups. Two-sided *p*-values less than 0.05 were considered statistically significant. We analysed survey data using IBM SPSS Statistics for Windows v27.0 (Armonk, NY, USA: IBM Corp). The interviews were digitally recorded and transcribed using MS Teams. Transcripts were reviewed for accuracy and then analysed for emergent themes using an inductive approach [30] with NVivo Version 12 (QSR International Pty Ltd., Burlington, MA, USA).

### 2.8. Ethical Approval

This study had ethical approval from the Macquarie University Human Research Ethics Committee (Project ID5480). The Inner West Council also approved this evaluation study. All participants in the intercept surveys and stakeholder interviews provided informed consent before joining the study.

## 3. Results

### 3.1. Pilot Play Street Attendance and Visitor Characteristics

Approximately 60 adults and children came to the Cary Street Play Street. The local organising committee set up tables where food and drinks were provided and shared amongst the Play Street visitors. We observed children engaging in a variety of activities, including, but not limited to, riding bicycles and scooters, playing ball games, throwing frisbees, blowing bubbles, jumping in puddles, drawing with chalk, and general free play and running around (see Appendix A).

A total of 33 adults were invited to complete the survey with 32 agreeing to participate (response rate: 97%). Of the 32 survey respondents (Table 1), the majority were female (77%), aged 35 to 54 years old (88%), lived in the same postcode as the pilot Play Street (97%), and visited in groups of adults plus children (88%). Among the adults who came with children (n = 23), 66% had primary school-aged children and 44% had children aged 5 years or younger. The average duration of the visit was 100 min (SD = 40.8 min) and ranged from 8 min to 2 h.

### 3.2. Reasons for Visiting the Play Street

Out of the 32 survey responses, 72% of respondents said they came to socialise and meet up with friends, while half said they came to meet new people (50%) or to let their children play (53%). Around one fifth said they came to check out something new (22%), that the Play Street sounded interesting (22%), and the location was convenient and close to home (22%). About one third of respondents heard about the Play Street by word of mouth from neighbours (31%). The other main sources of information about the event were flyers via letterbox drop (19%) and social media such as the Council’s Facebook page (19%).

### 3.3. Community Experiences and Perspectives of the Pilot Play Street

We asked visitors to rate the Play Street in terms of safety, organisation, and enjoyment (Table 2). The event was rated very positively overall with 97% of respondents rating it as “safe or very safe”. There were no differences between respondents’ perceptions of safety when stratified by demographic characteristics. 

The pilot Play Street was rated as “well or very well organised” by 81% of respondents, overall. On average, respondents with children aged 5 years or younger had lower ratings of organisation compared to people with older children (mean = 4.10 vs. mean = 4.85; t = −2.52, *p* = 0.027). The respondents with children in primary school rated the event better organised than those without children in primary school (mean = 4.57 vs. mean = 4.00; t = 3.51, *p* = 0.002). The respondents in groups with 1–2 children rated the event as better organised than those who came in groups with more than 2 children (mean = 4.8 vs. mean = 4.0; t = 2.30, *p* = 0.009). Respondents with previous experience of playing in the street did not differ significantly in their rating of the pilot Play Street’s organisation compared with those without previous experience (mean = 4.82 vs. mean = 4.25; t = −2.04, *p* = 0.058).

All respondents rated the pilot Play Street as “enjoyable or very enjoyable”. When asked about their children’s enjoyment of the Play Street, male respondents rated the event higher on children’s enjoyment than female respondents (mean = 5.00 vs. mean = 4.67; t = 2.38, *p* = 0.029) as did respondents with children in high school (mean = 5.00 ± 0) compared to those without children in high school (mean = 5.00 vs. mean = 4.67; t = 2.38, *p* = 0.029).

We also examined whether the street was a usual play space for children and what the respondents’ children would be doing if they were not attending the pilot Play Street event at that time. For the majority of children (48%), parents/caregivers said the pilot Street Play Street was the first time they had been permitted to play in the street; 60% of parents reported that their child or children would be engaging in screen-based or other sedentary activities if they had not been at the Play Street that day; 24% said their child or children would be playing at a playground or park.

When asked “If the Council decides to hold more Play Streets in the future (here or in other locations), how often do you think they should do this?”, 63% of respondents indicated a future Play Street should be held a few times a year, and 28% said 1–2 times per month would be appropriate.

### 3.4. Stakeholder Perspectives of Play Streets

Due to COVID-19 lockdown in Sydney implemented the week after the pilot Play Street, there was a very low participation rate in the stakeholder interviews. The final sample consisted of n = 2 stakeholders, both of whom were employees of different local governments with responsibilities related to community recreation and activation, including exploring the feasibility of implementing Play Streets. The small sample size precluded thematic analysis and we present the details herein based on the broad structure of the interview guide.

#### 3.4.1. Factors Contributing to the Success of the Pilot Play Street or Future Play Streets

The stakeholders credited the extensive consultation process with residents in the local community at the pilot Play Street that led to final arrangements being satisfactory to all parties (e.g., number of events planned, points of road closures). They also highlighted the importance of having a group of local residents who were active supporters of the initiative and willing to organise it. The local champions were also open to the views of other community members against Play Streets and were prepared to compromise.

#### 3.4.2. Challenges

The stakeholders were asked if there were any challenges they encountered or expect to encounter when organising Play Streets. Both stakeholders cited the need to balance competing views about Play Streets as the primary challenge, echoing the key factor for success which was being willing to compromise to reach a consensus. Issues around public liability, insurance, and safety (e.g., concerns from traffic committee) were also noted. One stakeholder commented that Play Streets would be more suited to areas where houses tend to be on very small blocks, with little yard space, and are, therefore, less relevant for area where residences are on larger blocks of land with space for front or back yards.

#### 3.4.3. Other Insights and Learnings

Both stakeholders emphasised that Council’s role is as facilitator and partner, rather than as instigator of Play Streets. The Councils’ role is to make it easy with process and resources for community members to hold their own Play Street. They will help locals organise and with the process but ultimately the community should own the event and make it happen themselves. The Councils’ role is to ensure that the processes are in place to make it easy and manageable for the community to run Play Streets in safe and appropriate way, such as making resources and support available, guidance on navigating Council website, how to get approval from traffic committee for road closure, and information regarding any fees (if applicable). Both stakeholders highlighted the need for sustainable and long-term focus; Play Streets should not be one-off initiatives.

## 4. Discussion

This evaluation shows that a pilot Play Street in the Inner West area of Sydney was feasible and acceptable to the community and that streets can be used as safe and enjoyable outdoor recreation spaces for children in higher density urban neighbourhoods in an Australian metropolitan context. Our results are consistent with evidence about Play Streets conducted in other countries, such as the USA [21,31], UK [32], Chile [18], and Belgium [19]. We found that the pilot Play Street reached appropriate community segments (i.e., parents and carers of children) from the neighbourhood, and children engaged in a variety of free play activities. The pilot Play Street was rated highly in terms of safety, enjoyment, and organisation, indicating promise for wider dissemination in the Inner West area of Sydney in the future.

The Inner West Sydney pilot Play Street demonstrated key elements of best practices for successful implementation [23,24]. The pilot Play Street had support from the local council throughout the planning process, including assistance with navigating council bureaucracy, funding permits, and providing barriers for street closure. Local residents and organisers championed the Play Street and were willing to negotiate with dissenting neighbours in their street and reach a compromise acceptable to both sides. This included reducing the number and duration of pilot Play Street events and reducing the length of the section of street to be closed off to traffic. This commitment to compromise is notable, since three out of four initially selected streets withdrew from the process due to push back from neighbours against Play Streets.

To the best of our knowledge, this is the first Play Streets evaluation in Australia reported in the peer-reviewed literature. However, a larger movement, 1000 Play Streets (https://www.playaustralia.org.au/1000-play-streets, accessed on 2 November 2022), launched in 2019 in four Australian states (Victoria, South Australia, Western Australia, and Queensland) different to the state in which this pilot was conducted (New South Wales). The 1000 Play Streets evaluation findings reported in the grey literature indicated a high level of support from participants for children to play on the streets more regularly in residential areas and the belief in the positive impacts of Play Streets on people’s access to opportunities for outdoor recreation space and their wellbeing, as well as strengthening community social capacity, reflecting the peer-reviewed evidence base [33,34].

In contrast to the other Play Streets, the Inner West Sydney pilot Play Street was smaller in scale, approximately 200 m in length and a one-off event, compared to other multi-site initiatives held on multiple occasions [18,19,20,21,23]. The pilot Play Street also did not provide any play equipment or organise activities for visitors. For example, in a multi-site Play Streets initiative in Ghent, Belgium [19], the council provided a box of play equipment to Play Streets local organisers, made larger equipment such as trampolines and jumping castles available for hire, and supported one organised event involving a circus school visit to each Play Street location. In comparison, the Inner West Sydney pilot Play Street was a “bring your own” event and children led their own play, including riding scooters, skateboards, and bicycles; playing ball sports like soccer and cricket; and general unstructured play, such as running around, drawing with chalk, and blowing bubbles. However, the pilot Play Street is located in a postcode with a higher proportion of residents with tertiary education and higher median income than state and national averages [35], whereas Play Streets in other countries have targeted neighbourhoods with lower socioeconomic status and underserved communities [18,20,21].

Several contextual factors highlighted the impacts of climate change and the need for human adaptation to ensure future sustainability of Play Streets in the Australian context. The pilot Play Street’s implementation was delayed due to conditions arising from a severe bushfire season in the Australian summer of 2019/2020. Poor air quality (smoke and ash from bush fires) and extreme heat led to the event being postponed from late January 2020 to March 2020. On the day of the event in March 2020, there was heavy rain which may have impacted the visitor turnout. Additionally, Sydney entered COVID-19 lockdown in the week following the pilot Play Street and this had negative impacts on the response rates for stakeholder interviews.

Future considerations and adaptions for holding Play Streets in the Australian context include monitoring temperature, UV, and air quality (air pollution, pollens) especially in the warmer months and holding Play Streets during times when these factors are within safe levels, such as playing during cooler times of day. Play Streets organisers could also provide water spray or sprinklers for hot day play and arrange to have shade and sunscreen available. At the local government level, councils could support Play Streets organisers with provision or funding of shade options in the short term and develop a longer term shade growth plan by planting more trees where possible. Australia is also currently experiencing La Niña weather patterns with above average rainfall and flooding in many parts of the country. During wet weather, Play Streets organisers could offer parents and caregivers wet weather play options like jumping in puddles and reminding them to provide children with appropriate rain gear. Local authorities should ensure rain damaged pavements and road surfaces are repaired promptly. If community movement restrictions such as those arising from COVID-19 are introduced in the future, supporting neighbourhoods to have impromptu street play to alleviate social isolation and build community resilience should be considered. [36]

The key strength of this study is the process evaluation which identified key issues relevant to further implementation of Play Streets in an urban Australian context. The prominence of climate change and COVID-19 related factors was unexpected and has not been reported in previous Play Streets research. Nonetheless, this evaluation was limited by the post-only design which did not assess impacts of the pilot Play Street on children’s physical activity and other health and wellbeing indicators. However, this formative research aimed to determine the feasibility and acceptability of the pilot Play Street rather than to evaluate its impact on health-related indicators. Future implementation and impact evaluation of Play Streets in Sydney could include indicators used by the large-scale 1000 Play Streets initiative for better comparability across states. Future evaluations should also include assessment of environmental factors like UV, temperature, and air quality which will influence wider implementation of Play Streets in Australia.

## 5. Conclusions

Overall, this study demonstrated that the Play Streets initiative is feasible and acceptable in an urban Australian setting. The sustainability of Play Streets in the future will be enhanced by successful navigation of competing views about Play Streets among community members, and by effective planning and adaptation to extreme conditions arising from climate change. Play Streets organisers will need to account for conditions like extreme heat, high UV levels, and poor air quality when holding events to ensure a safe and enjoyable experience for children and community members.

## Figures and Tables

**Table 1 ijerph-20-02476-t001:** Pilot Play Street intercept survey respondent characteristics (n = 32).

Characteristic		n	%
Gender	Female	6	19.4
	Male	24	77.4
	Other, preferred not to say	1	3.2
Age group (years)	25–34	1	3.1
	35–44	17	53.1
	45–54	11	34.4
	55–64	2	6.3
	≥65	1	3.1
Postcode of residence	Same as pilot Play Street	31	96.9
	Outside Inner West	1	3.1
Group size	1 (adult only)	4	12.5
	2	4	12.5
	3	5	15.6
	4	8	25
	5	5	15.6
	≥6	6	18.7
Group composition	Adults only	9	28.1
	Adults + children	23	71.2
Age of children *	≤5 years	10	43.5
	Primary school-aged	21	65.6
	High school-aged	5	21.7

* Only in respondents who reported having child(ren) with them (n = 23), percentages sum to >100% as more than one category could be selected.

**Table 2 ijerph-20-02476-t002:** Perceptions of pilot Play Street by demographic characteristics (n = 30) ^#^.

Demographic	Safety Mean (SD)	df	t	95% CI	*p* Value	Organisation Mean (SD)	df	t	95% CI	*p* Value	Enjoyment Mean (SD)	df	t	95% CI	*p* Value	Children’s Enjoyment * Mean (SD)	df	t	95% CI	*p* Value
**Gender**																				
Male (n = 6)	4.83 (0.41)	28	0	−0.439; 0.439	1	4.67 (0.52)	28	1.034	−0.409; 1.242	0.31	4.50 (0.55)	28	−0.739	−0.629; 0.295	0.466	5.00 (0)	17	2.38	−0.439; 0.439	0.029
Female (n = 24)	4.83 (0.48)					4.25 (0.94)					4.67 (0.48)					4.67 (0.59)				
**Age Group**																				
<45 years (n = 16)	4.75 (0.58)	25	0.268	−5.13; 0.156	0.28	4.38 (0.81)	28	0.272	−0.584; 0.763	0.788	4.69 (0.48)	28	0.64	−0.255; 0.487	0.527	4.67 (0.62)	21	−0.875	−0.703; 0.287	0.391
≥45 years (n = 14)	4.93 (0.27)					4.29 (0.99)					4.57 (0.51)					4.88 (0.35)				
**With Children Aged** **≤** **5 years ***																				
Yes (n = 10)	4.70 (0.68)	9	−1.406	−0.783; 0.183	0.193	4.10 (0.88)	11.6	−2.522	−1.393; −0.099	0.027	4.50 (0.53)	15.6	−1.761	−0.764; 0.071	0.098	4.70 (0.68)	21	−0.298	−0.522; 0.414	0.769
No (n = 13)	5.00 (0)					4.85 (0.38)					4.85 (0.38)					4.77 (0.44)				
**With Children in Primary School ***																				
Yes (n = 21)	4.86 (0.48)	21	−0.414	−0.861; 0.575	0.683	4.57 (0.75)	20	3.508	0.232; 0.911	0.002	4.71 (0.46)	21	0.607	−0.520; 0.949	0.301	4.71 (0.56)	21	−0.706	−1.128; 0.556	0.488
No (n = 2)	5.00 (0)					4.00 (0)					4.5 (0.71)					5.00 (0)				
**With Children in High School ***																				
Yes (n = 5)	5.00 (0)	21	0.712	−0.320; 0.653	0.712	4.80 (0.45)	11.7	1.306	−0.239; 0.950	0.217	5.00 (0)	17	3.289	0.139; 0.638	0.004	5.00 (0)	17	2.38	0.038; 0.629	0.029
No (n = 18)	4.83 (0.51)					4.44 (0.78)					4.61 (0.50)					4.67 (0.59)				
**Play in the Street ***																				
The first time (n = 12)	4.75 (0.62)	11	−1.393	−0.645; 0.145	0.191	4.25 (0.87)	15.9	−2.043	−1.158; 0.022	0.058	4.58 (0.52)	20.6	−1.221	−0.635; 0.166	0.236	4.75 (0.62)	21	0.98	−0.458; 0.503	0.923
Had previous experience (n = 11)	5.00 (0)					4.82 (0.41)					4.82 (0.41)					4.73 (0.47)				
**Opportunity to Play Outside ***																				
Yes (n = 21)	4.86 (0.48)	21	−0.414	−0.861; 0.575	0.683	4.57 (0.68)	21	1.06	−0.550; 1.692	0.186	4.71 (0.46)	21	0.607	−0.520; 0.949	0.551	4.81 (0.40)	1	0.806	−11.50; 13.11	0.566
No (n = 2)	5.00 (0)					4.00 (1.41)					4.50 (0.71)					4.00 (1.41)				
**Group Size: Adult**																				
1 (n = 6)	4.67 (0.82)	5.2	−0.849	−1.141; 0.570	0.433	4.17 (1.33)	6.1	−0.376	−1.605; 1.177	0.72	4.5 (0.55)	25	−0.962	−0.673; 0.244	0.345	5.00 (0)	17	2.38	0.038; 0.629	0.029
>1 (n = 21)	4.95 (0.22)					4.38 (0.81)					4.71 (0.46)					4.67 (0.59)				
**Group Size: Children ***																				
Small (1–2) (n = 15)	5 (0)	7	1.426	−0.247; 0.997	0.197	4.8 (0.41)	8.5	2.3	0.224; 1.376	0.009	4.73 (0.46)	21	0.52	−0.327; 0.544	0.108	4.80 (0.41)	21	0.73	−0.323; 0.673	0.473
Big (>2) (n = 8)	4.63 (0.74)					4.0 (0.93)					4.63 (0.52)					4.63 (0.74)				
**Group Size: Adults + Children ***																				
Small (2–4) (n = 14)	4.86 (0.54)	21	−0.159	−0.448; 0.384	0.875	4.64 (0.63)	21	0.991	−0.340; 0.959	0.333	4.64 (0.50)	21	−0.663	−0.558; 0.289	0.515	4.86 (0.36)	10.6	1.156	−0.275; 0.878	0.273
Big (>4) (n = 9)	4.89 (0.33)					4.33 (0.87)					4.78 (0.44)					4.56 (0.73)				

# Analytic sample n = 30, two responses were excluded from analyses because one respondent identified as ‘other’ for their gender and another one did not respond to the gender question. * Only included participants with children (n = 23).

## Data Availability

The data presented in this study are available on request from the corresponding author. The data are not publicly available due to privacy restrictions.

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
