# Peer review of "Navigating Neighbourhood Opposition and Climate Change: Feasibility and Acceptability of a Play Street Pilot in Sydney, Australia"

_ijerph, 2023, doi:10.3390/ijerph20032476_

Round 1
Reviewer 1 Report
The paper is well written. The analysis can be stronger. For example, demographic data could be used to determine how different populations viewed the event.
My suggestions to improve this manuscript:
* be clear about the criteria being evaluated up front. The article read more like a summary of findings versus a research paper. What questions were you seeking to answer and what data was used to answer those questions?
* on a related note, the data presented could be more precise given the amount of demographic data collected. With some data, response were broken down by demographic variables which provided more clarity as to who was responding in certain ways and why. Again, this should be guided by the a priori criteria set.
* With more structure in the method section, I believe the introduction section should include info to support the a priori themes identified.
Best
Reviewer 2 Report
An informative article about a sympathetic initiative to improve the quality of life in the city for children. The research design is a nice mix of quantitative and qualitative research.
The phenomenon could theoretically be placed in a historical line of urban planning, with attention to the playing child in the city. An example is the significant contribution of architect Aldo Van Eyck in Amsterdam. Rather than a relationship with sustainability, one could think of the relationship with this tradition, the geographic theory of Third Space (Soja, etc.) and, for example, the creation of intermediate areas in Barcelona. But that is beyond the scope of this article, which is primarily an evaluation of the Play Street phenomenon.
I had assumed that the article was indeed a pragmatic evaluation of the play streets phenomenon and had no further high scientific pretensions. I like those kinds of articles and found this article publishable in that regard. Nevertheless, here are some considerations, which may lead the editors to decide to make some adjustments to the article. I hope this is satisfactory and will consider my further position as a reviewer.
(1, 3) The main questions of the article can be found in the conclusions, perhaps these could be explicitly included in the beginning of the article as a question section, with the main question: How can play streets contribute to increasing the child-friendliness of urban environments, especially considering on the non-permanent, incidental character of a play street and the necessary involvement and organizational strength among residents. Is this a sustainable phenomenon? How is the Play Street phenomenon appreciated by those involved? The value of the article lies in the scientific evaluation of the Play Street phenomenon, in particular the organizational background and the appreciation of those involved.
(2) Play streets are a relatively new phenomenon in the effort to make the built environment more child-friendly. The article contributes to highlighting and evaluating this phenomenon. Unfortunately, the article does not provide any insight into the longer tradition of designing the built environment for (playing) children and relation to public health. Maybe this is a forgotten chapter in urban planning? Or does the play street-phenomenon fit into theoretical visions of the city and the participation of its residents? For example, the ideas of Soja about Thirdspace? The article now has a limited objective: a pragmatic evaluation and is in that sense adequate and publishable. But the outline of some context could benefit the subject.
(4) The mix of quantitative and qualitative research is positive. Due to circumstances, the stakeholder analysis is very limited, but nevertheless informative. Follow-up research could use methods of so-called performative research. Perhaps in this article the methodology can be extended in that direction by adding photo material of the play street in question, so that a more vivid picture of the phenomenon is created.
(5) It may be considered to include clearer questions at the beginning of the article (see 1). The conclusions in section 5 are rather general, tending towards a general description of the phenomenon of play streets, which is better included as an introduction. The question is whether attention to climate conditions could not be better limited.
(6) References are appropriate given the limited purpose of the article as a pragmatic evaluation.
Reviewer 3 Report
The article submitted for review is characterized by a logical presentation of the material, it is well structured, the available tables make it possible to better understand the research results. The purpose, tasks and conclusions correspond to the title of the article and are clearly related to each other. Positively evaluating the results of the scientists' research, the following recommendations should be given to the authors in order to improve it:
1. In the abstract, clearly state the purpose of the article and determine the practical significance of the conducted research.
2. Add keywords to the article.
3. It is advisable to separate the literature review section and not to combine it with the introduction section, where the background of the research is usually investigated, relevance is justified, problems and prospects of the research direction are considered, and the purpose and tasks of the article are noted.
4. In the literature section, special attention should be paid to sustainable development, namely its ecological component and children's awareness of the importance of careful use and environmental protection. At the same time, we recommend that you pay attention to and add to the reference the following important studies by scientists:
Marhasova, V., Tulchynska, S., Popelo, O., Garafonova, O., Yaroshenko, I., Semykhulyna, I. (2022). Modeling the harmony of economic development of regions in the context of sustainable development. International Journal of Sustainable Development and Planning, Vol. 17, No. 2, pp. 441-448.
Kosach I., Duka A., Starchenko G., Myhaylovska O., Zhavoronok A. Socioeconomic viability of public management in the context of European integration processes. Administratie si Management Public. 2020. Vol. 35. P. 139-152.
Anishchenko, V., Marhasova, V., Fedorenko, A., Puzyrov, M., Ivankov, O. (2019). Ensuring environmental safety via waste management. Journal of Security and Sustainability Issues, 8(3): 507–519.
5. In the research, it is recommended to justify the practical significance of the mentioned proposals and developments, to provide information about the interested parties in the results of this research and to indicate for whom they can be useful in practical activities.
6. In the conclusions, it is advisable to provide more detailed information about the obtained research results.
7. It is also recommended to note in the conclusions the novelty of the conducted research, in contrast to existing research on this topic.
In general, I believe that the article can be accepted and published in a scientific journal after taking into account the specified recommendations.
